# A Novel Spider Monkey Optimization for Reliable Data Dissemination in VANETs Based on Machine Learning

**DOI:** 10.3390/s24072334

**Published:** 2024-04-06

**Authors:** Deepak Gupta, Rakesh Rathi

**Affiliations:** Department of Computer Science and Engineering, Government Engineering College Ajmer, Ajmer 305001, India; rakeshrathi@ecajmer.ac.in

**Keywords:** VANET, spider monkey, leader, machine learning, optimization, relay, NS2.35

## Abstract

The growth in linked and autonomous vehicles has led to the emergence of vehicular ad hoc networks (VANETs) as a means to enhance road safety, traffic efficiency, and passenger comfort. However, VANETs face challenges in facilitating trustworthiness and high-quality services due to communication delays caused by traffic, dynamic topology changes, variable speeds, and other influencing factors. Hence, there is a need for a reliable data dissemination scheme capable of reducing communication delays among hops by identifying effective forwarder nodes. In this paper, we propose a novel, weighted, estimated, spider monkey-based, nature-inspired optimization (w-SMNO) method to generate a set of efficient relays. Additionally, we introduce a dynamic weight assignment and configuration model to enhance system accuracy using a neural network based on backpropagation with gradient descent optimization techniques to minimize errors in the machine learning model. The w-SMNO also incorporates a distinct algorithm for effective relay selection among multiple monkey spider groups. The simulation results demonstrate substantial improvements in w-SMNO, with a 35.7% increase in coverage, a 41.2% reduction in the end-to-end delay, a 36.4% improvement in the message delivery rate, and a 38.4% decrease in the collision rate compared to the state-of-the-art approaches.

## 1. Introduction

Vehicular ad hoc networks (VANETs) have emerged as a transformative technology with the combination of connected and autonomous vehicles. These networks play an important role in transforming road safety, optimizing traffic flow, and enhancing the passenger experience [1,2,3]. However, data dissemination in VANETs involves challenges related to optimizing the dissemination process, ensuring reliable data delivery, minimizing latency, and utilizing available resources efficiently [4]. These aspects often involve searching for optimal routes, managing communication overhead, dealing with intermittent connectivity, and ensuring the dissemination of information among vehicles in a timely and efficient manner [5,6]. The complexity of these tasks places VANET data dissemination problems within the realm of NP-hard problems due to their computational challenges and the need for efficient solutions to solve them optimally. One of the critical hurdles facing VANETs is the communication delay resulting from factors such as traffic congestion and frequent alterations in network topology. Traffic congestion may lead to a broadcast storm that results in a high collision rate of packets and increases delays [7]. These delays can impede the effectiveness of applications crucial to vehicular safety and traffic management. To address this challenge, a robust and reliable data dissemination scheme is imperative, capable of minimizing communication delays and identifying optimal paths or the effective relay for an efficient and fast information exchange among vehicles. Figure 1 illustrates the fundamental communication scenario within a vehicular ad hoc network (VANET), where any incident detected via a vehicle is promptly relayed to nearby vehicles within its transmission range. Additionally, a relay identification process is implemented within the network to transmit received information to further hops, thus mitigating the potential for a broadcast storm. Furthermore, using mobile devices, laptops, roadside units (RSUs), or other network-connected devices, the event data are disseminated to other internet-connected services and societal resources.

Swarm intelligence algorithms, such as particle swarm optimization (PSO), ant colony optimization (ACO), and genetic algorithms (GA), among others, have been explored for various optimization problems in VANETs, including routing, data dissemination, and resource allocation [8,9,10,11]. These algorithms, inspired by the collective behavior of social insects or other animals, provide innovative solutions to enhance the performance of VANETs. Nature-inspired optimization algorithms combine features from monkeys and spiders for VANET (vehicular ad hoc network) problem-solving [12]. The spider monkey algorithm (SMA) stands as a metaheuristic algorithm drawing inspiration from the foraging behavior of monkeys in search of food. This innovative approach orchestrates a collective of agents, represented as ‘monkeys’, operating collaboratively to explore and navigate through a defined search space with the aim of uncovering the most optimal solution [13]. A trust-based, cluster-oriented approach, SCSF covers the aspect of node isolation techniques from mobility-based ad hoc networks to solve real-life optimization issues that are NP-hard problems as well [14].

The primary motivation for this research is the necessity of providing vehicle networks with a reliable and efficient dissemination of data by reducing the delay in communications and the broadcast storm effect so that the related services can be available to end users via these types of environments on time. By utilizing the nature-inspired framework of spider monkey optimization theory and a comprehensive strategy based on the selection of information forwarders from a group of vehicles, our work not only tackles complex technical challenges but also offers a novel solution to the current problems.

In this article, we introduce an advanced approach that goes beyond conventional solutions to enhance data dissemination in VANETs. Our proposed method, the weighted, estimated, spider monkey-based, nature-inspired optimization (w-SMNO), integrates the concepts of local and global leader selection to strategically identify efficient relay nodes. This innovative, hybrid approach ensures a comprehensive solution to the unique challenges posed by VANETs. The local leader selection process empowers individual subgroups to autonomously identify and nominate potential relay candidates, raising adaptability in dynamic environments. Simultaneously, the global leader plays an important role in overseeing the entire network, optimizing relay node selection. We additionally examine the effects of the monkey population and the number of local groups by incorporating the limits. By setting specific thresholds for population sizes and local leaders, the system can dynamically adjust its strategies based on the number of active nodes, ensuring optimal relay node selection and effective data dissemination even under varying traffic conditions. This technique offers the flexibility needed to accommodate fluctuations in network density, optimizing the balance between communication efficiency and network resource utilization. As the population size surpasses or falls below predefined thresholds, w-SMNO adapts its relay node selection mechanisms, thereby ensuring an adaptive response to the network. The w-SMNO methodology was meticulously designed to address the unique challenges of VANETs, providing a comprehensive solution to improve coverage, reduce end-to-end delays, enhance message delivery rates, and minimize collision risks. Through extensive simulations, we demonstrate the substantial advantages of w-SMNO over state-of-the-art methods, showcasing its efficacy in elevating the performance and reliability of VANET data dissemination systems. The information distribution in ad hoc networks is also reflective of the security situation in a number of ways, specifically in relation to potential threats from negative nodes. Further improving this feature will help make the ad hoc vehicle communication system better overall [15]. Figure 2 illustrates the targeted idea behind the proposed method in relation to the vehicular network.

The following are the contributions made in this article:Problem formulation: This study formulated the data dissemination problem in a novel and innovative way by presenting it as a multi-objective optimization problem. The forwarding mechanism configurations define the solution space, and factors like coverage, the message delivery rate, the end-to-end delay, and the collision rate are included in the goal.Solution framework: This research introduces a novel framework for data dissemination, utilizing optimization based on the weighted, estimated, spider monkey-based, nature-inspired optimization (w-SMNO) method, establishing an efficient set of relay nodes through a hybrid approach that combines local and global leader selection strategies.Learning model: The study provides a machine learning technique to enhance accuracy and minimize errors in dynamic weight prediction using backpropagation with the gradient descent method. Moreover, the model includes the development and implementation of various novel adaptive optimized algorithms inspired by spider monkey behavior and expressions for dynamic weight assignment and configuration.Performance efficacy: This research evaluated the performance of w-SMNO using numerous simulation tests for different networking metrics and probability functions. Further, the study included a comparison of experimental results with other state-of-the-art approaches.

The rest of this manuscript is structured as follows: Section 2 provides a summary of the existing literature and the related research in this domain. The methodology and guiding principles of the suggested approach are elaborated upon in Section 3. A thorough analysis is outlined in Section 4, with an emphasis on how well the suggested approach performs in relation to various decision metrics. In the end, Section 5 provides the conclusion and possible directions for further investigation.

## 2. Related Works

The literature contains several works in the development of data dissemination approaches in VANET for optimization. In the work of [16], multi-objective optimization such as the adaptive jumping multi-objective firefly algorithm (AJ-MOFA) was designed to eliminate hazardous conditions due to congestion or a broadcast storm using clustering. Further, the approach was enhanced for a forwarding mechanism that uses probabilistic forwarding. However, data dissemination in urgent situations such as vehicle clashing, traffic jams or long queue at toll stations was not taken into account. Also, the selection of parameters dynamically for the meta-heuristic was lacking in the proposed method. In the work [17], the issue of a secure path to disseminate the information in a vehicular network through trusted fuzzy logic routing schemes for smart cities using a candidature-based path selection approach is addressed. Nevertheless, the complexity of the proposed solution escalates due to the incorporation of multiple methodologies within a unified framework. In [18], particle optimization was employed for efficient data dissemination. A network was established with vehicles acting as nodes to promptly transmit emergency messages using the FIFO and time delay-based multipath routing (TMR) method. Particle swarm optimization (PSO) was applied for the optimal selection of paths. The analysis of the results incorporated metrics such as the delay, throughput, packet loss ratio, and energy consumption. Additionally, it is worth noting that, in the majority of instances, the approach involved the consideration of stationary nodes, which represent a non-realistic behavior. Through [19] the scenario explored the event of a network failure at a specific vehicle when other vehicles might assume control to relay information to the required nodes, thereby ensuring continued network performance. The document delves into routing schemes founded on several bio-inspired approaches. Furthermore, it elaborates on the significance of nature-inspired algorithms, elucidating their contributions to addressing diverse issues within ad hoc networks. According to  [20], nature-inspired algorithms represent a potent toolset for addressing optimization challenges. Various algorithms falling under this category have been explored due to their relevance in distinct facets of the Internet of Vehicles (IoV), with particular emphasis on security, routing, and parking space management. Through their thorough investigation, it is evident that the application of nature-inspired algorithms has the potential to enhance and optimize the overall performance of IoV networks. In [21], the resolution to minimize the charging delay to mobiles was efficiently tackled through the utilization of a mobile assignment problem considering IoT-enabled WSN via mobile chargers in a service network. An approach [22] inspired by the Salp swarm was put forth to effectively locate vehicular nodes in the NLOS area. Due to the interference and obstructions in this area, vehicle localization is difficult and has not been successful in the non-line of sight (NLOS) region. Neighborhood awareness and delay metrics demonstrated enhanced performance. Nevertheless, the method ignores the effect of node mobility and assumes that all nodes have the same transmission power. A network traffic prediction model based on machine learning considering the random forest method was proposed for traffic from the combination of V2V and V2R networks [23]. The work [24], addressed concerns related to network scalability and optimal route identification in VANETs; the grasshopper optimization-based node-clustering algorithm was employed to optimize the selection of cluster heads, particularly in scenarios with variable node density. This innovative approach successfully mitigates network overhead in situations with unpredictable node density. Nevertheless, it falls short in accounting for latency and other real-time variations in traffic-related scenarios. Energy replenishment issues in mobiles through a rechargeable sensor network covering the balanced consumption of energy and the separation of multiple redundant mobile sensors is addressed in [25], and a solution is proposed through novel balancing and sensor dispatch approaches For the first issue, an energy balancing algorithm is proposed that uses cascaded movement to improve the cascading schedule. For the second issue, a redundant mobile sensor dispatch algorithm is proposed that prioritizes the mobile sensors most in need of energy replenishment for replacement via a charged and calibrated redundant mobile sensor.

In [26], the protocol introduced is the fuzzy bacterial optimization zone-based routing, designed to efficiently determine a brief and stable communication route within confined environments. Similarly, In [27], utilizing the gray wolf optimization technique, introduced a clustering algorithm for dissemination, reducing the overhead of intricate networks. A different approach based on a swarm [28] was suggested for VANET route optimization proposals through the ant colony optimization (ACO) technique. The performance resulted in a reduced number of cluster heads (CHs) in various network scenarios compared with multi-objective particle swarm optimization (MOPSO) and comprehensive learning particle swarm optimization (CLPSO) [29]. Moreover, sensitivity to environmental factors not considered to overcome integration challenges. For stable clustering in VANETs, an insightful, novel, naive Bayesian likelihood-based method of estimation for traffic distribution was proposed (ANTSC) [30] based on a traditional artificial intelligence methodology and a situation where performance in a variety of network scenarios was compared in terms of traffic flow [31] and traffic management [32] using clustering; the proposed method was more effective. In [33], a method to disseminate data based on the packet delivery ratio was proposed for a 6th-generation vehicular network for the selection of information forwarder vehicles responsible for reducing the delay in transmission.

For the purpose of determining the best routes in VANETs, a clustering routing approach based on swarm optimization (CRBP) was put forth; it consists of cluster formation, path computing, and the ideal route for efficient dissemination as key components in [34]. In order to minimize frequent path disconnection through periodically exchanging signals to maintain the most recent trusted values, the optimized node selection routing protocol (ONSRP) [35] was proposed for the optimized node selection procedure to choose the most optimized vehicle. In [36], a genetic whale optimization algorithm (GWOA) was combined with an improved cognitive tree protocol for routing (MCTRP) to manage path exploration, identifying channels for the route, and data communication. All the other nodes join as child nodes to form a tree, with the node with the greatest node identity opted for as the root node. By determining the cost, a suitable path is selected using GWOA, and the node with the lowest cost is selected as the relay for transmission.

Routing protocols’ performance issues related to an urban scenario using a clustering approach based on density peaks and particle swarm optimization (PSO) is covered in [37]. However, the approach is limited to a basic traffic model. In [38], an artificial intelligence-driven, software-defined network (SDN) controller proposed a centralized routing scheme with mobility prediction for VANET. This predictive capability enables the accurate estimation of the successful transmission probability and average delay of each vehicle’s request amid frequent changes in network topology. This estimation is particularly useful in a stochastic urban traffic model, where vehicle arrivals follow a non-homogeneous Poisson process. However, the latency and other issues resulting from real-time variations in traffic scenarios were not taken into consideration. The use of machine learning algorithms in vehicular networks addresses issues related to latency, ensuring the smooth flow of traffic and enhancing road safety, which is addressed in [39]. These algorithms were applied within the context of real-time VANET scenarios to assess their feasibility and effectiveness. Additionally, this approach considers future directions and challenges associated with implementing machine learning techniques. Utilizing machine learning in Vehicle-to-Infrastructure (V2I) communication within VANETs involves employing software comprising both static and mobile agent approaches. Decision tree and Q-Learning algorithms are utilized to detect potential serious incidents, aiming to enhance bandwidth, PDR, and E2E delay [40]. In [41], a method was proposed for selecting the optimal relaying set during the broadcast procedure. This approach employs supervised learning to classify candidate nodes based on various factors, followed by sorting them according to relay quality to select the best ones. Moreover, the method assumes perfect channel knowledge for all nodes and overlooks the effects of node failures on dissemination efficiency. The issues of coverage to enhance the receiving information probability at multiple receiver units were covered in [42] and resolved through the theory of entropy.

To the best of our knowledge, the group- and subgroup-based relay selection that is specifically based on a machine learning model was utilized for the first time in the VANET environment using our suggested method as a dissemination protocol. In addition, we suggested creating threshold-oriented subgroups based on the size of the monkey population for scenarios that are grounded in reality.

## 3. Proposed w-SMNO

Monkey optimization (MO) is a novel swarm intelligence algorithm inspired by the intelligent foraging behaviors of spider monkeys. These monkeys exhibit a social structure in which they form smaller groups within larger ones, dynamically merging or separating based on food availability. MO effectively manages premature convergence and stagnation, potentially leading to improved solutions. Spider monkeys assemble in groups led by a senior female member who acts as the primary decision-maker. To optimize foraging, large groups split into smaller clusters, each led by a local female leader overseeing the foraging paths. The following outlines the food procurement process of spider monkeys:As spider monkeys begin searching for food, they measure the distances to available food sources. Using these distance measurements, the monkeys adjust their positions within the group and recalibrate the distances as needed.The position of the local leader is updated within the group. If this update does not occur for a set number of iterations, the members within that group redirect their search for food sources in different directions.Eventually, the main leader of the largest group maintains its best position. If stagnation occurs, the main leader creates subgroups.

Figure 3, shows the conceptual steps for MO.

Optimization operations for the spider monkey population include the following sub-operations:

**Initialization**: Initialize a population of spider monkeys within a search space and assign locations and directions to these spider monkeys, representing potential solutions to the optimized relay search problem.

**Evaluation**: Evaluate the fitness or quality of each monkey’s position/solution using the objective function of the problem.

**Movement**: Similar to moving vehicles, to find the best node to forward the information, spider monkeys mimic their social behavior by implementing movement rules. They swing between trees to simulate the exploration and exploitation of the search process via the random exploration of new solutions within the search space or the transmission range, while exploitation involves moving towards promising solutions based on the previously found best solutions.

**Update**: Update the positions of the monkeys based on their movement strategies that improve the fitness of the solution and maintain a balance between both exploration and exploitation for a comprehensive search.

**Termination**: This includes the achievement of a satisfactory solution via selection to optimize the information forwarder node.

The proposed w-SMNO operates through the collective behavior of spider monkeys exploring and exploiting the search space to find optimal solutions. The application of SMOA can be enhanced to optimize data dissemination strategies in VANETs to efficiently distribute information among vehicles in the network in order to enhance the efficiency of message delivery, reduce the communication overhead, minimize delays, and ensure reliable information sharing among vehicles. The challenges include limited network coverage, a highly dynamic network topology due to vehicle mobility, intermittent connectivity, and varying network conditions. Additionally, it can aid in the allocation of resources such as bandwidth or transmission power for effective data dissemination while minimizing the communication overhead. The working steps explained as follows for the proposed SMNO are presented in Figure 4.

Step 1: Initialization of population

Vehicles equivalent to a spider monkey population of size N are the neighbor vehicles moving in different directions, and they are considered initially within the transmission range of the information sender vehicle. Such a one-hop population is inspired by the swarm intelligence approach.

Step 2: Global leader selection (GLS)

In this phase, one of the population members based on parameter evaluation is elected as the global leader. The global leader (GL) is updated with the best-fit node to create the subgroups through greedy selection. This leader is questionable for the fission of the population to search for the relay node or food through the creation of subgroups of the population. After the finalization of the best relay node in later steps, the global leader fuses all the subgroups through fusion.

Step 3: Local leader selection (LLS)

In the LLP, the global leader vehicle assigns the task of searching for the best information forwarder node, equivalent to sufficient food in the spider monkey method, to multiple local leaders (LLs) by creating subgroups of one-hop nodes from the experience of the local leader and group members. With the help of greedy selection, the local leader is updated as the best-fitting spider monkey or node in that local group. The fitness values of the newly identified forwarder node are calculated.

Step 4: Local relay identification

The local leader monkey or node locally updates the old forwarder details selected with the new ones when it attains a new, higher fitness value. Relay or forwarder nodes through local leader subgroups are identified.

Step 5: Global relay identification

From the set of relay nodes identified through the local leader in step 4, the global leader chooses the best fit candidate based on the computation of defined parameters for efficient dissemination.

Step 6: Relay finalization

The best relay nodes are elected to disseminate the information to other hop.

The search space is explored as the group members update their positions. There is a high perturbation in the initial iterations that later is reduced gradually. The optimization of the vehicle population corresponding to the spider monkeys is represented in the equations below. These equations represent the movements of nodes within search space; updating positions based on movement rules and the evaluation of fitness for the best candidate selection on a greedy basis, including speed and direction, supports guidance, representing the monkeys’ exploration and exploitation.

The proposed method ensures an even distribution of the available population (*N*) of monkeys. In the variable population (*G*), { G≥3 | ∈3,4,5,…N } is considered in the approach to achieve maximum real-time coverage. Each monkey specification comprises *Q* directional elements as variables in identifying adequate food sources (relay nodes). The Monkey Distribution function ($MD_{\left(qn\right)}$) is initialized for the qth dimension for the nth monkey, generating uniformly distributed random numbers between 0 and 1, having upper and lower bounds (UB,LB), as shown in Equation (Equation 1). Subsequently, the monkeys’ positions are adjusted based on their previous locations only when the ensuing solution surpasses the preceding one, as specified in Equation (Equation 2). The leader’s ($SL_{I}$) position of the *J*th subgroup is then randomly chosen from the set I∈G and I≠q. Additionally, the positions of the monkeys and subgroup leaders, including the Principal (Global) leader ($PL_{q}$), in the *q*th dimension are updated using Equation (Equation 3).
(1)$$MD_{\left(qn\right)}=LB\left(MD_{q}\right)+R*\left[UB\left(MD_{q}\right)+LB\left(MD_{q}\right)\right]\;for\;R\in random\left[0,1\right]$$
where q∈Q and n∈G
(2)$$Next\left(MD_{\left(qn\right)}\right)=MD_{\left(qn\right)}+R*\left[SL_{Jq}-MD_{\left(qn\right)}\right]+R^{\prime }*\left[MD_{\left(In\right)}-MD_{\left(qn\right)}\right]$$
(3)$$Next\left(MD_{\left(qn\right)}\right)=MD_{\left(qn\right)}+R*\left[PL_{Jq}-MD_{\left(qn\right)}\right]+R^{\prime }*\left[MD_{\left(In\right)}-MD_{\left(qn\right)}\right]$$
where $R^{\prime }\in \left[-1,1\right]$

Considering the node location with the coordinates (X,Y,Z), ignoring *Z* for motion on the plane surface and ΔY for the vehicle lane change, the monkey’s corresponding position in Equation (4) for the kth vehicle in *t* time is shown below.
(4)$$X_{k}\left(t+1\right)=X_{k}\left(t\right)+\Delta X_{k}\left(t\right)$$
where $X_{k}\left(t+1\right)$ is the next position in (t+1) time, and $X_{k}\left(t\right)$ is the current position of the kth monkey in *t* time, respectively. $\Delta X_{k}\left(t\right)$ is the position variation to support parameters’ computation.

### 3.1. Local Leader Validation

For the evaluation of the best vehicle node from the population for the local leader, the corresponding function $F(X_{k})$ is defined to evaluate the node performance equivalent to the fitness function in the spider monkey optimization approach. The identification of the global best relay candidate by the global leader among the selected relays relative to the best monkey position is given in Equation (Equation 5), returning the optimized and maximum values among the nodes (monkeys).

At $X_{k}\left(t+1\right)$, optimize the condition for the best solution;
(5)$$candidate\leftarrow \mathrm{return}\{max\left(F\left(X_{k}\left(t+1\right)\right)\right)\}$$

The specific details and variations of the approach involve additional strategies or refinements to guide the vehicle (monkey) movements or based on the problem being solved. For the kth vehicle $V_{k}$ in the population (*N*), the optimized fitness function *F* evaluation is based on the considered nodes’ parameters, including the range of transmission ($R_{T}$), speed (*S*), energy consumption (Ec), throughput (Th), and neighbor size (*n*) to assess their performance, connectivity, or overall efficiency within the network while comparing for the leader. A list of all populations with the mentioned parameter values at any time instant, as shown in Figure 5, is maintained by the global leader. In this case, the source node is the global leader’s (GL) process to select local leaders for subgroups using the parameter value list.

Leader node selection is crucial for efficient communication and relay functionality. Threshold-based equations are commonly used to determine whether a node should act as a leader based on various parameters. These equations define conditions that must be met for a node to perform a specific action of a subgroup leader. The fitness evaluation to select LL for cut-off *C*, having a total dimension weight (*W*) for all *Q* dimensions, is given in Equations (Equation 6)–(8).
(6)F(X)>C;
for;
(7)$$F\left(X\right)=\sum_{k=1}^{N}\frac{w_{k}\left(R_{T}\right)+w_{k}\left(S\right)+w_{k}\left(Ec\right)+w_{k}\left(Th\right)+w_{k}\left(n\right)}{Total\;weight\;(parameters)}\cdot W$$
where;
(8)$$W=\int_{q=1}^{Q}\left[w{\left(R_{T}\right)}_{q}+w{\left(S\right)}_{q}+w{\left(Ec\right)}_{q}+w{\left(Th\right)}_{q}+w{\left(n\right)}_{q}\right]$$

For the selection of local leader nodes considering threshold or cut-off values, an equation that filters nodes according to criteria is based on the scenario where the set of parameters *P* is associated with each node, and the approach seeks to select some nodes that meet a specific cut-off value, *C*, while considering a dynamic maximum limit *L* to the number of selected nodes. To achieve this filtering of nodes based on the cut-off and limit including $P_{A}$ represents the value of parameter *P* for node *A* (where *A* ranges from 1 to *N*), *C* is the cut-off for parameter *P*, and *L* is the maximum limit of the selected leaders for subgroups. Ensuring that the nodes meeting the threshold value *C* are selected until the maximum limit, *L*, for local leaders is reached is shown in Algorithm 1.

**Algorithm 1:** LL Validation

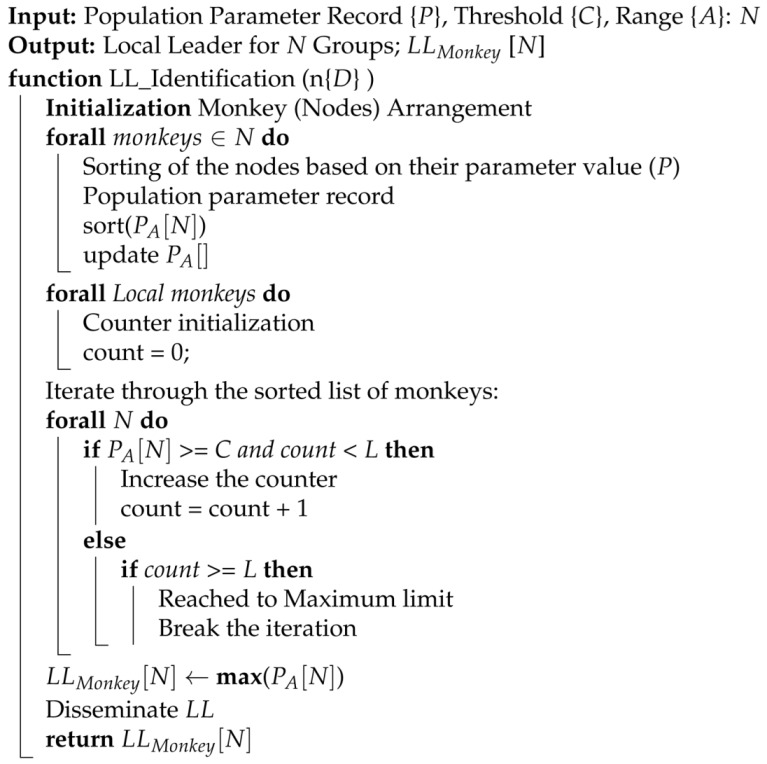



### 3.2. Groups’ Sub-Categorization

Subgroups’ member allocation for *N* vehicles after LL selection based on $P_{A}$ is determined through dynamic threshold-based allocation. $R_{TA},\;S_{A},\;Ec_{A},\;Th_{A},\;n_{A}$ are different cut-offs values for $P_{A}$. Nodes are assigned to groups based on which threshold $P_{A}$ meets. Algorithm 2 represents the members’ allocation to the *L* subgroups. $P_{AB}$ represents the value of parameter *B* for node *A* (where *A* ranges from 1 to *N*, and *B* ranges from 1 to *M*). $C_{BQ}$ represents the threshold value for parameter *B* and group *Q* (where *Q* ranges from 1 to the total number of groups, *L*).

**Algorithm 2:** Monkeys’ Assignment to Subgroups

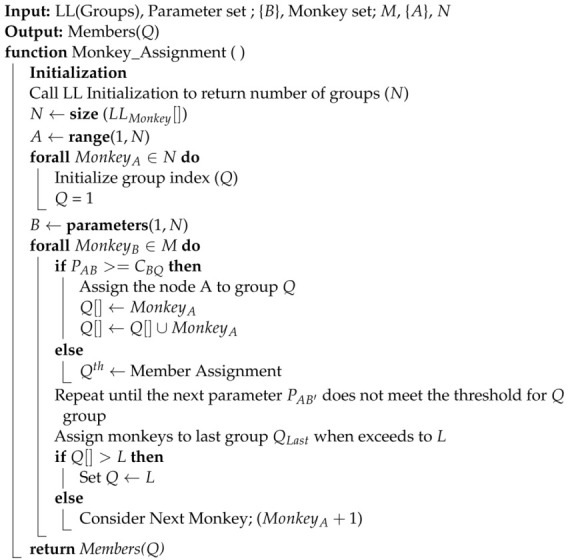



### 3.3. Relay Validation by LL

VANETs can be divided into subgroups, and within these subgroups, vehicles can dynamically elect or appoint local leaders to manage and optimize communication within their respective subsets. Local leaders play a crucial role in identifying relay nodes within their subgroups, optimizing the routing of messages and data. This is particularly beneficial in scenarios where communication needs to be extended beyond the direct communication range of individual vehicles. The process of selecting a relay node from subgroups entails identifying the most appropriate node based on a variety of factors or parameters that impact the selection of a relay node via LL. These are represented in Figure 6. The proposed approach includes the weight assignment for each parameter based on its importance or relevance to the selection process. For instance, the consumption of energy for a node might be more critical than speed in some scenarios.

The weight matrix (W[]) is defined as in Equation (Equation 9), representing the dynamic weight (*w* for each parameter $R_{T},S,Ec,Th,n$).
(9)$$W\left[\right]=\left\{\begin{array}{c} w_{RT}\\ w_{S}\\ w_{Ec}\\ w_{Th}\\ w_{n} \end{array}\right\}:\forall w\in range\left(0,1\right);$$

### 3.4. Weight Configuration Learning Model

The proposed approach utilizes the machine learning technique to dynamically provide the weights for parameters. In this method of learning weights’ values, a neural network architecture consists of input, hidden, and output layers for which weights are learned during the training process. These weights represent the parameters that the network adjusts to make predictions or learn patterns from the input data. The representation of a neural network defines basic layers for input and output along with the hidden layer. The input layer receives the input features of vehicles, such as the direction, speed, neighbors, and others. Each node in this layer represents a feature for a node as the input data. The computation for weight configuration is performed via hidden intermediate layers between the input and output layers. Each node in a hidden layer performs a weighted sum of inputs and applies an activation function. Finally, the output layer produces the final output of the network. The values of weights in a neural network are initialized randomly at the beginning of training and are updated throughout the training process to improve the network’s ability to make accurate predictions. These updates are performed using the gradients of the loss function with respect to the weights. For dynamic values of weights, during the training phase of the neural network, these weights are adjusted iteratively using backpropagation with gradient descent optimization techniques to minimize the error or loss between the predicted and actual outputs. The vehicles begin to analyze the data to identify patterns and develop logic according to the pattern, after which a leader is chosen. Nearly 80% of the data were entered into the learning process to find the dynamic weight values’ moving patterns. The previous observations were kept. The testing procedure was then run with 20% of the data currently being received because the training set discovered absolute patterns through a large number of inputs collected via the simulation, as shown in Figure 7.

In the backpropagation with gradient descent is a key algorithm used to train neural networks by adjusting the weights and biases to minimize the error between predicted and actual outputs. Below is the explanation for the method with the involved equations in the process of an updated weight.

Forward propagation: During forward propagation, the input data (*I*) are passed through the network to make predictions (*p*) for biases (*b*) in the predicted output ($O^{\prime }$) for the actual output (*O*). In our method, we consider a simple neural network with one hidden layer. $D^{\left[1\right]}$ and $D^{\left[2\right]}$ are weights of the hidden and output layers, respectively, for any activation function (α). Equations (10) to (13) represent the forward propagation for the weight of the dynamic prediction output through the activated hidden layer.
(10)$$p^{\left[1\right]}=D^{\left[1\right]}\cdot I+b^{\left[1\right]}$$
(11)$$a^{\left[1\right]}=\alpha \cdot \left(p^{\left[1\right]}\right)$$
(12)$$p^{\left[2\right]}=D^{\left[2\right]}\cdot a^{\left[1\right]}+b^{\left[2\right]}$$
(13)$$O^{\prime [1]}=\alpha \cdot p^{\left[2\right]}$$

Calculating loss: The difference between the predicted output, O’, and the actual output, *O*, is computed using a mean squared error loss function (*G*) for learning loss computation and defined below in Equation (Equation 14).
(14)$$Reduction\leftarrow G(O^{\prime },O)$$

Backpropagation (gradient computation): Backpropagation involves calculating the gradients of the loss function with respect to the weights and biases, which are used to update the parameters, as shown in Equations (15) and (16). Using gradient descent, the gradients are computed by taking partial derivatives of the loss function with respect to each parameter.
(15)$$\frac{\delta (Reduction)}{\delta D^{\left[2\right]}}=\frac{\delta G}{\delta O^{\prime }}\cdot \frac{\delta O^{\prime }}{\delta p^{\left[2\right]}}\cdot \frac{\delta p^{\left[2\right]}}{\delta D^{\left[2\right]}}$$
(16)$$\frac{\delta (Reduction)}{\delta b^{\left[2\right]}}=\frac{\delta G}{\delta O^{\prime }}\cdot \frac{\delta O^{\prime }}{\delta p^{\left[2\right]}}\cdot \frac{\delta p^{\left[2\right]}}{\delta b^{\left[2\right]}}$$

Similar calculations are performed for $D^{\left[1\right]}$ and $b^{\left[1\right]}$ by backpropagating the error through the network, as shown below in Equations (17) and (18).
(17)$$\frac{\delta (Reduction)}{\delta D^{\left[1\right]}}=\frac{\delta G}{\delta O^{\prime }}\cdot \frac{\delta O^{\prime }}{\delta p^{\left[1\right]}}\cdot \frac{\delta p^{\left[1\right]}}{\delta D^{\left[1\right]}}$$
(18)$$\frac{\delta (Reduction)}{\delta b^{\left[1\right]}}=\frac{\delta G}{\delta O^{\prime }}\cdot \frac{\delta O^{\prime }}{\delta p^{\left[1\right]}}\cdot \frac{\delta p^{\left[1\right]}}{\delta b^{\left[1\right]}}$$

Weight update: In Equations (19) to (22), the weights and biases are updated using the computed gradients scaled with a learning rate (β) to take a step towards minimizing the loss.
(19)$$D^{\left[2\right]}=D^{\left[2\right]}-\beta \cdot \frac{\delta (Reduction)}{\delta D^{\left[2\right]}}$$
(20)$$b^{\left[2\right]}=b^{\left[2\right]}-\beta \cdot \frac{\delta (Reduction)}{\delta b^{\left[2\right]}}$$
(21)$$D^{\left[1\right]}=D^{\left[1\right]}-\beta \cdot \frac{\delta (Reduction)}{\delta D^{\left[1\right]}}$$
(22)$$b^{\left[1\right]}=b^{\left[1\right]}-\beta \cdot \frac{\delta (Reduction)}{\delta b^{\left[1\right]}}$$

Using the above-mentioned equations by gradient decent method, the core of the backpropagation algorithm is optimized. They are used iteratively across multiple epochs to update the parameters until convergence, reducing the loss and improving the model’s performance with the training data. Adjusting the learning rate (β) is crucial to control the step size during weight updates and prevent overshooting or slow convergence.

### 3.5. Monkeys’ Weights in LL for Relay Identification

Minimum and maximum values are determined from the detected data to normalize and save them in a table. After the deviations have been updated, the observation is noted in the state table following the node movement. Then, a value between 0 and 1 is assigned to the original weight and bias values. The values between two successive observations are computed for positive and negative values. The neural network is created to determine the accuracy across layers. The input receives and sends the neighbor vehicle parameters to the network. The hidden layer depends on the action of the network; it will be increased, and there may be one or more than one. The hidden layer is in charge of precise dispensation in neural networks. They carry out several tasks simultaneously, including data conversion, feature generation, etc. The final output layer sends the results of the given problem. During the process, the stages where neuron known as the activation function is utilized. For this, the neural network uses weights to mix various inputs. The activation is gathered at the distance between the neuron’s center and the input. The method decides whether the neuron state is activated through the weighted sum and bias values. According to the output layer, the data transmitter chooses the leader, which then initiates the data transmission in a broadcasting fashion. The leader is selected via optimum calculations, thus remaining stable until the transmission is completed. For each node in a particular LL group among $N^{\prime }$, a score based on the parameters and their weights is computed, considering the values of parameters (*P*) and the respective dynamic weight (*W*). Score (Sc) computation for any node *A* in $LL_{Q}$ is shown below from Equations (23) to (25).
(23)$$Sc_{A}=P_{A}\cdot W$$
or
(24)$$Sc_{A}=\left[R_{T_{A}}S_{A}Ec_{A}Th_{A}n_{A}\right]\cdot \left[\begin{array}{c} w_{RT}\\ w_{S}\\ w_{Ec}\\ w_{Th}\\ w_{n} \end{array}\right]$$
or
(25)$$Sc_{A}=R_{T_{A}}w_{R_{T}}+S_{A}w_{S}+Ec_{A}w_{Ec}+Th_{A}w_{Th}+n_{A}w_{n}$$The score generation and relay selection from the Qth group of a local leader (LL) having $N^{\prime }$ nodes is determined via Equations (26) and (27), respectively.
(26)$$Sc_{A}=\sum_{i=1}^{Population(LL)}{\left(P_{A}\right)}_{i}\cdot W_{i}$$
(27)$$Relay\left(LL_{Q}\right)\leftarrow max\left(Sc_{1},Sc_{2},\ldots Sc_{A},\ldots Sc_{N^{\prime }}\right)$$

### 3.6. Relay Node Identification via GL

The aggregation of relays from all *Q* groups creates the candidate set to choose the best node via the global leader (GL). Algorithm 3, as given below, iterates through all nodes and evaluates their scores computed via local leaders using the scoring method by keeping track of the maximum score encountered and the node associated with that score. After an iteration through all the nodes, the node with the highest score represents the node with the maximum score. The candidates participating in a relay through LL based on the score achieved are determined through Equations (28)–(30).
(28)$$CandidateSet=[Relay_{1},Relay_{2},Relay_{3}\ldots ,Relay_{Q}]$$
(29)$$S_{Candidate}=\left[S_{Relay_{1}},S_{Relay_{2}},S_{Relay_{3}}\ldots ,S_{Relay_{Q}}\right]$$
(30)$$Relay\left(GL\right)\leftarrow maxScore\left(S_{Candidate}\right)$$

**Algorithm 3:** Relay Identification via Global Leader (GL)

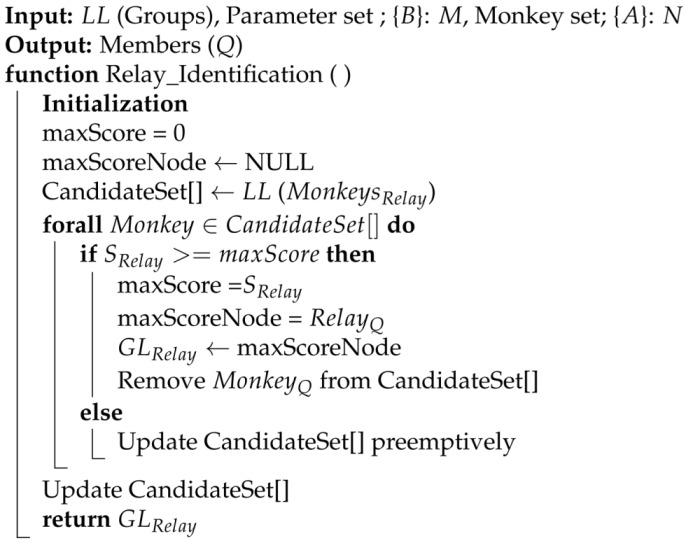



### 3.7. Threshold Instances for w-SMNO

In the process of selecting relays for our proposed system, some cases and subcases must be meticulously considered both for local groups and via the global leader to ensure optimal performance and reliability. A careful analysis of the relay’s sensitivity and response time is crucial to guarantee its suitability for dissemination and routine operating scenarios. Different cases and subcases for relay selection through w-SMNO in VANET ensure optimal performance in dynamic vehicular environments. In some cases, it would be highly beneficial to select the LL with the next highest weighted score instead of the highest score. Let us consider two scenarios involving the mentioned case, one with the maximum score and the other with the next-to-maximum score, based on the population size in local groups and the number of neighbors to the local leader. For these scenarios, threshold is defined for factors, as shown below in Table 1. When both the population of the specific group *Q* is $P_{LL_{Q}}$ and the number of neighbors in that respective group, $N_{LL_{Q}}$, is greater than the population and local leader threshold, then the maximum weighted group using the dynamic parameter weight node from the learning model is considered to enhance the dissemination as the condition is to maintain a balance between groups and a leader, according to the population size. Similarly, if both the number of neighbors in that respective group, $N_{{LL}_{Q}}$, is less than the threshold ($LL_{Threshold}$) and the population of *Q* subgroups, $P_{{LL}_{Q}}$, is also less than the population threshold ($P_{Threshold}$), then the next maximum weighted group, $Q^{\prime }$, is considered. For other conditions, the maximum weighted group is selected. An assumption has been made for the minimum neighbors’ local leaders ($LL_{Threshold}$) with actual neighbors’ local leaders ($N_{LL}$) and the dynamic population of a specific group ($P_{LL}$) with a population threshold ($P_{Threshold}$) for an efficient dissemination process, where $LL_{Q}$ and $LL_{Q}^{\prime }$ are, respectively, the maximum score and the next maximum score of local groups’ leaders.

Figure 8 illustrates a dynamic model depicting the threshold considerations for both local leaders’ thresholds and each group population within a given total population size. The graph showcases the intricate relationship between these two entities as they navigate the challenges and opportunities inherent to their respective roles. The limitation to the number of local leaders for relay consideration demonstrates a distinct threshold that signifies the point at which their influence becomes the most impactful within their immediate community or region. Simultaneously, the graph represents the threshold for the group population at large, emphasizing the critical juncture at which the collective strength and cohesion of the entire population come into play. This figure serves as a valuable tool for understanding the nuanced interplay between local leaders and the broader group population, offering insights into the optimal points of engagement and coordination to maximize positive outcomes within the larger framework.

## 4. w-SMNO: Performance Analysis

The following subsections consist the experimental evaluation for performance analysis of proposed approach including the establishment of simulated vehicles’ indicating monkey population, their communication infrastructure, the network topology, and the implementation of related protocols and algorithms. Furthermore, specific parameters and metrics are delineated to evaluate the method’s effectiveness and performance within the simulation setting with an aim to enhance efficient dissemination through relay identification in VANET communication.

### 4.1. Simulation Organization

Simulation was conducted to demonstrate the effectiveness of the proposed w-SMNO, over specifically based estimation scheme (SES) [43], adaptive intersection selection mechanism using ant colony optimization (AISM) [44], and the adaptive jumping multi-objective firefly algorithm (AJ-MOFA) [16]. A comparative analysis of these are presented in Table 2.

For extensive experiments, NS2.35 served as the simulation platform. We employed the 500 m × 500 m heavy traffic area along Jaipur JLN Marg, India, sourced from OpenStreetMap, as depicted below in Figure 9, to generate a node density trace file using SUMO and MOVE. This file captures realistic mobility environments, including information on each node’s speed and location, which are then utilized via NS2.35 as a dataset to assess the performance of w-SMNO. In each simulation, the generated population set is divided into local groups ranging from 5 to 40, each containing populations ranging from 30 to 60 nodes. These groups, each containing populations, are introduced at specific time intervals to assess performance metrics, as explained in later sections. Nodes with speeds ranging from 20 to 100 km/h are included in the population to demonstrate the correlation between the results and a comparative analysis of UDP transport with CBR traffic, with a maximum flow rate of 100 vehicles. Table 3 provides an overview of the simulation parameters utilized in this approach.

The effectiveness of the w-SMNO approach when compared to the non-grouped conventional method to variable subgroups (number of LL) is demonstrated in Figure 10. The outcomes for various metrics are presented for a predetermined number of local groups. As local groups and the vehicle density increase, the effectiveness of w-SMNO undergoes an improvement when contrasted with non-grouped communication. This is because there is less repetition of similar information among nodes in the same group. Breaking down the population into multiple groups helps simplify the process of identifying relay nodes, making it less complex by handling fewer nodes within each group. The results further illustrate that the efficiency of the approach may fluctuate with the rise in the number of groups more than the limitation relative to the vehicle density or population, potentially affecting the overall performance. Table 4 outlines the multiple considered conditions for the evaluation and illustration of the same.

### 4.2. Evaluation Metrics

In this section, we outline the evaluation criteria employed to appraise the effectiveness and efficiency of the proposed system in comparison to alternative techniques. The selection of parameters significantly influences the efficacy of the solution for information dissemination through relays. Enhancing the overall performance entails a comprehensive understanding of parameters pertaining to population, groups, subgroups, relay selection, and other relevant factors. To optimize the performance, a systematic literature review was conducted in order to identify recommended values or ranges for similar parameters in both nature-based and conventional dissemination models.

Moreover, the proposed method’s performance was gauged across diverse network sizes to evaluate its scalability, ensuring that the parameters can adeptly adjust to varying node numbers while upholding performance standards. We evaluated the effectiveness of the suggested w-SMNO approach based on network coverage, message delivery proportion, end-to-end delay, and collision rate variables using the specifications provided for parameter configuration. Network coverage is a crucial metric to evaluate the effectiveness of communication coverage. A higher network coverage percentage signifies a more extensive and effective communication reach within the vehicular network, which is essential for the success of applications such as safety warnings, traffic management, and other cooperative services that rely on timely and reliable communication between vehicles. The message delivery proportion assesses the effectiveness of message dissemination and represents the percentage of messages that are successfully delivered to their intended recipients. A higher value is desirable, as it reflects the efficiency and reliability of the communication system. The end-to-end delay measures the time taken for a data packet to travel from the source vehicle to the destination vehicle, and it represents the total delay affecting the packet during its transmission through the network. Minimizing the end-to-end delay is essential to ensure timely communication. The collision rate measures the frequency of collisions between transmitted packets. It can occur when multiple vehicles attempt to transmit data simultaneously on the same communication channel, leading to signal interference and a loss of information. A lower collision rate is desirable for an efficient and reliable communication environment. All of these different metrics concentrate on assessing performance in terms of the total *n* local group each having nodes, *N*, for the total *P* population in *r* tests for the LL subgroup. The following provides an in-depth explanation of the metrics employed to assess performance.

Network coverage (ω): Network coverage refers to the extent of the geographical area within which communication is established between vehicles and infrastructure nodes. ω is the percentage ratio of the number of vehicles successfully reached or connected to the total number of vehicles in the network. Equation (Equation 31) defines the average ω.
(31)$$\omega_{Avg}=[\left(\sum_{test=1}^{r}\left(\int_{1}^{LL}\left\{\int_{1}^{n}\frac{N_{Linked}}{N_{LL}}\right\}*\frac{1}{P}\right)/r\right]*100$$

Message delivery proportion (μ): The definition of the message delivery proportion is the proportion of messages that, out of all the messages sent over the network, reach their intended recipients successfully. The average of μ for $M_{R}each$ messages in a subgroup from the total *M* messages is determined through Equation (Equation 32).
(32)$$\mu_{Avg}=[\left(\sum_{test=1}^{r}\left(\int_{1}^{LL}\left[\int_{1}^{n}\frac{M_{Reach}}{M_{LL}}\right]/M\right)/r\right]*100$$

End-to-end delay (τ): The end-to-end delay is defined as the total time it takes for a data packet to travel from the source to the destination in a communication network. It encompasses the transmission delay (TD), propagation delay (PD), queuing delay (QD), and processing delay (PrD). For defined factors, including the population and subgroups’ size, the average delay (τ) is shown in Equation (Equation 33).
(33)$$\tau_{Avg}=\left[\sum_{test=1}^{r}\left[\int_{1}^{LL}\left[\int_{1}^{n}\left(TD+PD+QD+PrD\right)*\frac{1}{r}\right]\right]\right]$$

Collision rate (σ): The collision rate refers to the proportion of collision occurrences in a communication system, particularly in scenarios where multiple devices contend for access to a shared communication medium. For the collision count (CC), among the total number of transmission attempts (*T*), the average of σ is expressed in Equation (Equation 34).
(34)$$\mu_{Avg}=\left[\left(\sum_{test=1}^{r}\left(\int_{1}^{LL}\left[\int_{1}^{n}\frac{CC}{T}\right]\right)*\frac{1}{r}\right)\right]$$

### 4.3. Simulation Outcomes

This subsection presented the simulation outcomes of this study, illustrating the efficacy of the proposed approach. With this methodology, numerous experiments were conducted, and the system’s performance was assessed across diverse scenarios through extensive simulations. To scrutinize the necessity of total local leaders (LL) in each subgroup for a random population, Experiment 1 was conducted, involving the variation in both LL and the population size over simulated time. Figure 11 demonstrates the analytical behavior of the experiment through a probabilistic approach for LL with a population size through a simulation. To the specific members limit, the number of subgroups are directly proportional to the monkey population. The analysis outcomes clearly represent the requirement for balance in LL with the increases in the population for the effective functioning and optimization of various systems, especially in scenarios involving large-scale networks. Careful consideration and adjustment of subgroup values become pivotal to achieving optimal performance and maintaining a harmonized operation as the number of nodes in the network grows.

Moreover, we assessed the stability of the proposed system by examining the various arrangements of both the minimum and maximum values for the LL and the population size in Experiment 2. Table 5 illustrates the groups with the combination of minimum and maximum values for LL and *P* for the cases of experiment. The results of the experiment are visually depicted in Figure 12.

While the stability of the system in networks with the maximum population and LL was relatively diminished compared to networks with the minimum LL and the maximum population, it is important to note that, concurrently, the overall performance of the w-SMNO method demonstrated a noteworthy enhancement. Opting for the minimum local leader (LL) value, coupled with the maximum population, proves advantageous when compared to other combinations. This choice minimizes the need for relay candidates, leading to a more streamlined and efficient system. The network exhibits increased stability, and the overall processing time is significantly reduced. With fewer relay candidates to manage, the system experiences enhanced efficiency and reliability, making the combination of the minimum LL and the maximum population a favorable configuration for optimal performance.

A comprehensive examination was undertaken through Experiment 3, concentrating on performance metrics like network coverage, the message delivery proportion, the end-to-end delay, and the collision rate as defined earlier to demonstrate the superiority of w-SMNO. This investigation involved a varying spider population (vehicles) and simulation time while maintaining a constant number of subgroups percentage (LL) to 5, 25, and 35 as an average of the minimum and maximum, with an initial population size of 50 nodes. Members of subgroups were allocated through the member allocation algorithm, as discussed in previous sections. The simulation result for the same is shown in Table 6 and Table 7.

Figure 13, Figure 14, Figure 15, Figure 16, Figure 17, Figure 18, Figure 19, Figure 20, Figure 21, Figure 22, Figure 23 and Figure 24 present the acquired results. The analysis reveals a remarkable enhancement in the reliability of the w-SMNO dissemination approach, manifested in a substantial decrease in the end-to-end delay and the rate of collision of packets alongside an augmentation in both network coverage and the message delivery rate to destinations. These improvements are notably superior to the existing state-of-the-art solutions. As the population size increases, the subgroups allocate members more effectively, thus improving the rate of message delivery, reducing the numbers of collisions, and enhancing the network coverage. Simultaneously, the early identification of relays further reducing the end-to-end delay. Overall it is promoting efficient data dissemination in vehicular communication across the entire network. These insights underscore the effectiveness of our approach, showcasing its potential for real-world deployments.

The behavior of the w-SMNO, as shown in Figure 13, Figure 14, Figure 15 and Figure 16, exhibited sufficient improvement in efficiency over time, especially concerning the defined key metrics of network coverage, the end-to-end delay, the message delivery rate, and the collision rate. As time progressed, the algorithm demonstrated a significant enhancement in network coverage, ensuring a wider span of effective communication. Similarly, the delay factor for the proposed method was reduced satisfactorily, as the time taken for both node processing due to local groups’ formation and the identification of an effective relay node was reduced. Finally, from the simulation outcomes for delays, it can be concluded that the time taken for information to traverse from a source to a destination showcases a favorable trend with time, indicating an optimized and expedited data dissemination process. Moreover, the message delivery rate exhibited a positive correlation with time, signifying an increasingly successful transmission of messages within the network. Simultaneously, the collision rate, representing instances of data packet conflicts, demonstrates a decreasing trend over time, underlining the algorithm’s adeptness in mitigating collisions and optimizing resource utilization. This efficient behavior of w-SMNO across these crucial metrics substantiates its efficacy and suitability for addressing the challenges inherent to network communication over varying temporal intervals.

Further, the competence of the w-SMNO method is notably pronounced, considering the similar metrics, particularly with the increase in the variable size of the spider monkey population, as shown in Figure 17, Figure 18, Figure 19 and Figure 20. As the population of spider monkeys grows, the algorithm exhibits an exceptional ability to adapt and optimize the network coverage. The expanded population contributes to a more extensive and robust network reach, ensuring that communication spans a larger area with increased efficiency. Simultaneously, the delay in the data demonstrates a positive trend with the growing and flexible size of the population. This is because, as the population rises, the sufficient number of local groups increases in accordance with the current population through the algorithm, and it shows its proficiency in minimizing delays, facilitating swift and efficient information dissemination within the w-SMNO network. Likewise, the message delivery rate exhibits a significant positive correlation with the enlarged spider monkey population. This is because of similar reasons involving multiple group formation with an appropriate size for each one. This indicates the algorithm’s capacity to handle larger populations adeptly, ensuring a higher success rate in delivering messages across the network, and the w-SMNO method excels in managing the complexities associated with an increased population size, contributing to an overall improvement in message delivery efficiency. In terms of the collision rate, it is reduced sufficiently. As we can see, there is an adequate increase in message delivery, indicating the reduction in the broadcast storm and efficiently handling the congestion, if any, due to heavy transmission. Related conclusions represent the instances of data packet conflicts; the algorithm consistently demonstrates a decreasing trend as the population expands. This shows the ability to mitigate collisions effectively via w-SMNO, optimizing resource utilization and enhancing the overall reliability and effectiveness in addressing the challenges posed in larger and more dynamic network environments.

## 5. Discussion

In this section, we delve into a comprehensive analysis that underscores the novelty of our work and clarifies the challenges in solving the delay problem for dissemination in VANETs through w-SMNO. Concurrently, we conduct a comparative study of the existing technologies, examining their impacts on transmission performance, including a consideration of the relay from the next hop without cluster formation [45], considering the challenge of a broadcast storm using a cluster-based forwarding mechanism [46], and route optimization for messages to their destination using particle swarm methods [47]. Our findings provide a notable reduction in delays, attributed to a substantial decrease in collisions and an enhanced success rate in message delivery facilitated via the implementation of the spider monkey technique. The proposed w-SMNO underwent a rigorous comparative analysis compared to existing methods, showcasing its superiority over both time and population in various performance metrics. As time advances, delays increase across all methods due to escalating communication overhead and collisions. Likewise, the message delivery ratio diminishes as the network congestion intensifies over time, resulting in packet drops. Figure 15 illustrates these delivery trends distinctly. At around 150 s, the packet drop rate peaks to alleviate congestion, followed by a subsequent rise in message delivery rates around 180 s, indicating congestion relief. Similar to other observations, it can be concluded that our approach outperforms others in reducing information delay, minimizing collisions, ensuring efficient message delivery, and extending network coverage over both with variable time and populations. w-SMNO strategically forms a sufficient number of local leaders within subgroups, adapting to variable population sizes. Thus it’s significantly improves service quality for vehicular ad hoc network (VANET) end users, providing accurate and timely information. The increase in the vehicular volume leads to a surge in information transmission, resulting in packet collisions that diminish both packet delivery rates and coverage. This, in turn, introduces delays as packets fail to reach their destinations. However, w-SMNO exhibits superior behavior and trends especially when considering the density of vehicles in subgroups and referencing local leaders (LLs). Figure 21, Figure 22, Figure 23 and Figure 24 illustrate these comparisons comprehensively. Strong challenges had to be overcome for the solution to work, including inconsistent speeds, frequent disconnections in sparse networks, and link stability in a dynamic topology. The novel aspect is how each subgroup’s relay vehicle selection procedure is created and executed with the help of w-SMNO. The weight model takes into account dynamic parameter values to accommodate real-time situations and handle coverage problems. Additionally, w-SMNO presents unique optimization strategies and scenarios designed to significantly lower the number of relay participants, reducing the communication overhead and addressing link stability issues in sparse and dynamic network environments. We observed significant improvements in coverage, message delivery, end-to-end delay, and collision rates, as demonstrated via our evaluation of various metrics. Thus, we can state with confidence that w-SMNO tackles the complexities of data distribution in VANETs and provides a thorough and well-thought-out answer to the problems under consideration.

## 6. Conclusions and Future Work

Problems such as effective data handling surface more frequently as ad hoc networks are used more and more. Overcoming these obstacles is essential to resolving issues like broadcast storms, delays, and collisions. We have presented a milestone strategy with our suggested method, which consists of grouping vehicles, determining sub-relays, and then choosing the best relay from each group, in comparing the included methods. With this novel approach, coverage and delay problems are successfully resolved using a population of spider monkeys. The design also integrates a neural approach to machine learning, which dynamically modifies parameters in real time, which is not utilized in existing mechanisms of dissemination. This technique uses the cooperative behavior of the spider monkey algorithm to act as a learning model for reconfiguration, encouraging the choice of a global relay node. The suggested model outperforms current algorithms in identifying local and global leaders. When the population is smaller, w-SMNO and our proposed algorithms demonstrated enhanced outcomes compared to the traditional ungrouped approach. Moreover, with dynamic member allocation, the proposed algorithms exhibit superior performance over other methods across varying population sizes. This study was dedicated to creating an efficient spider monkey-inspired algorithm to address the delay and collision challenges in data dissemination within VANETs. The primary goal was to enhance the performance, addressing the shortcomings observed in existing models. The results of the simulation at various time intervals indicate a significant improvement in VANET data dissemination performance, with improvements ranging from 35% to 45% for the metrics. This represents improvements in the V2V network’s general quality, trustworthiness, and dependability. In densely populated areas or situations with high message transmission rates, the decision factors taken into account regularly live up to expectations. The results highlight the accuracy of the suggested method in reducing broadcast storms and validating its dependability and robustness in dynamic environments. It is reasonable to conclude that the suggested method is dependable and efficient in light of these encouraging results. In the future, the proposed method can be enhanced for the analysis of security attacks, considering the compliance of spider monkey behavior with artificial intelligence techniques.

## Figures and Tables

**Figure 1 sensors-24-02334-f001:**
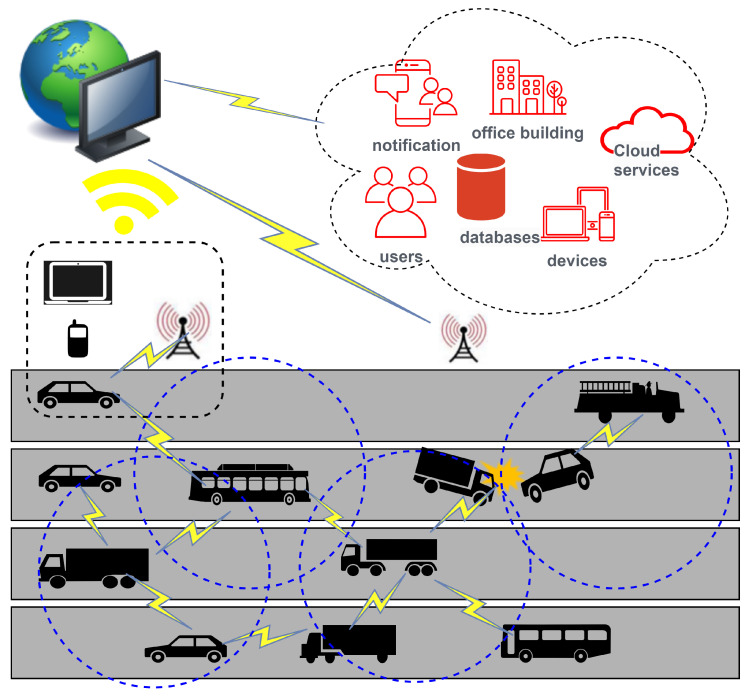
VANET communication environment.

**Figure 2 sensors-24-02334-f002:**
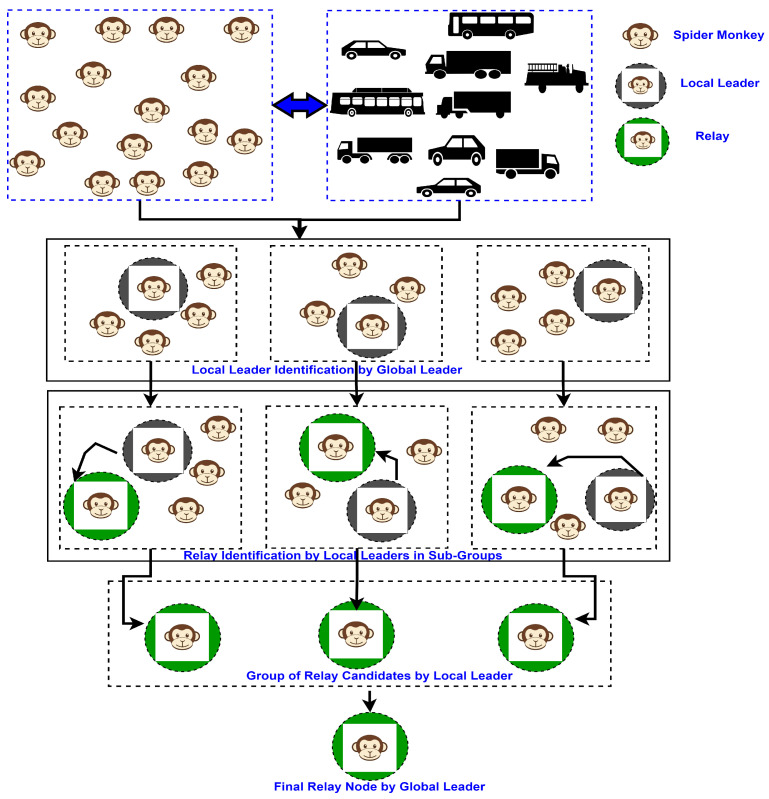
Spider Monkey communication model.

**Figure 3 sensors-24-02334-f003:**
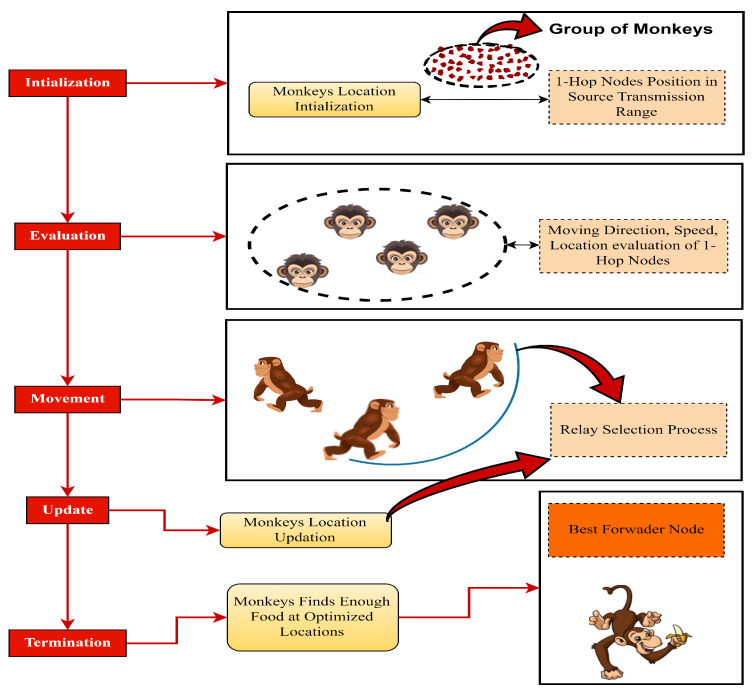
MO flow procedure.

**Figure 4 sensors-24-02334-f004:**
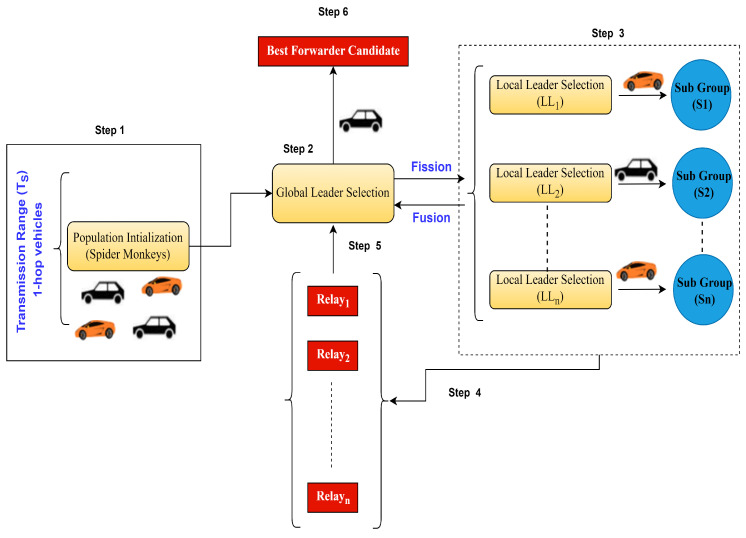
SMNO working steps.

**Figure 5 sensors-24-02334-f005:**
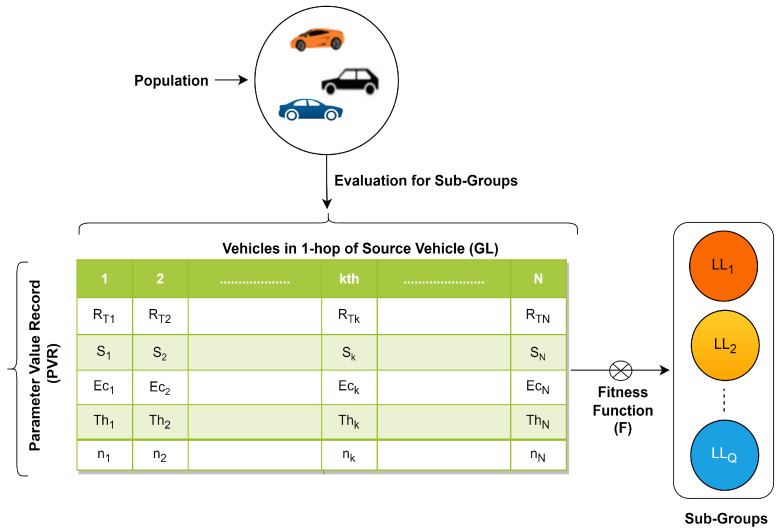
Population parameter record.

**Figure 6 sensors-24-02334-f006:**
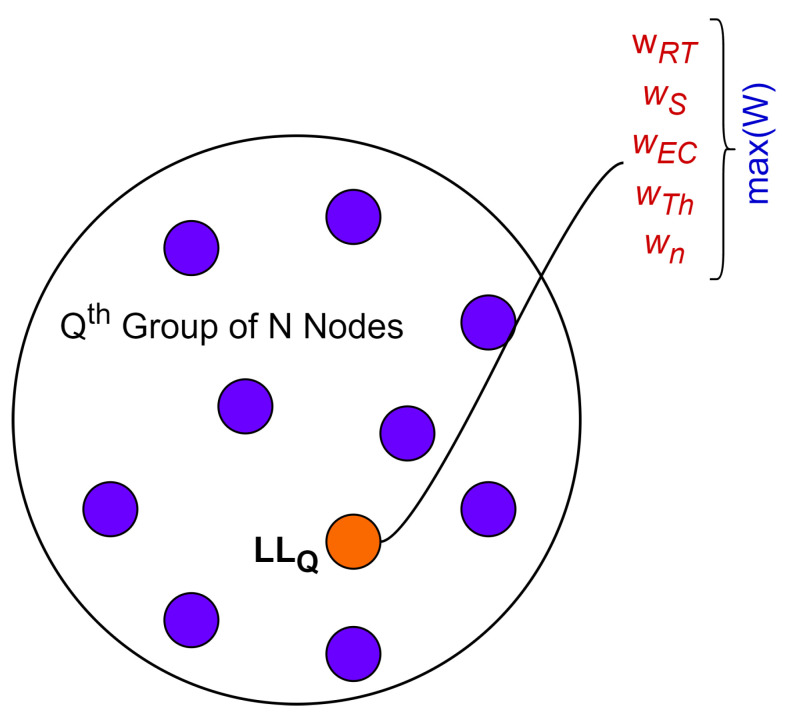
Qth LL group.

**Figure 7 sensors-24-02334-f007:**
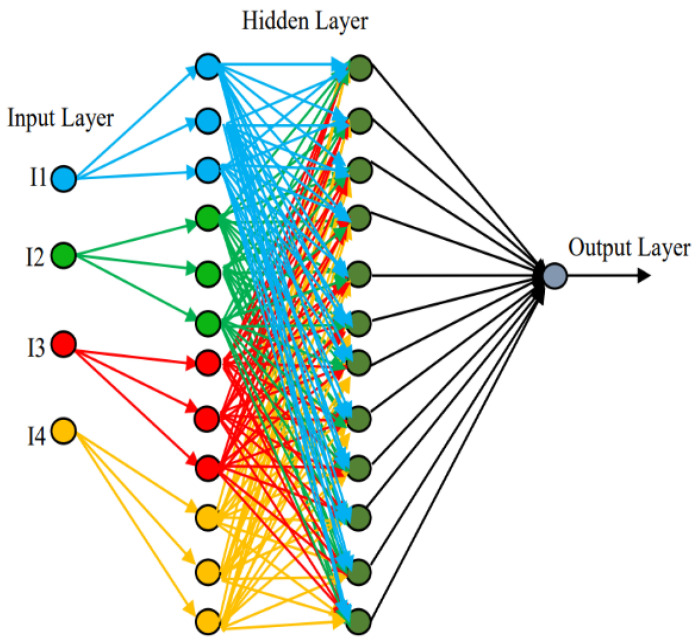
Weight learning neural network model for parameters.

**Figure 8 sensors-24-02334-f008:**
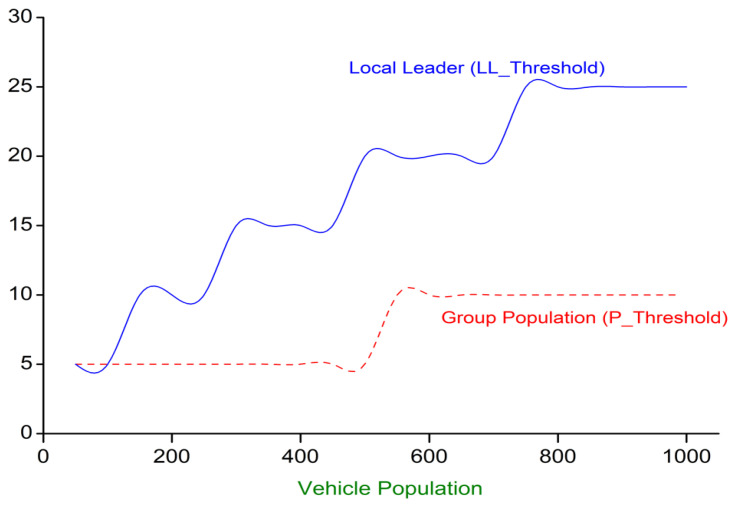
Threshold instances.

**Figure 9 sensors-24-02334-f009:**
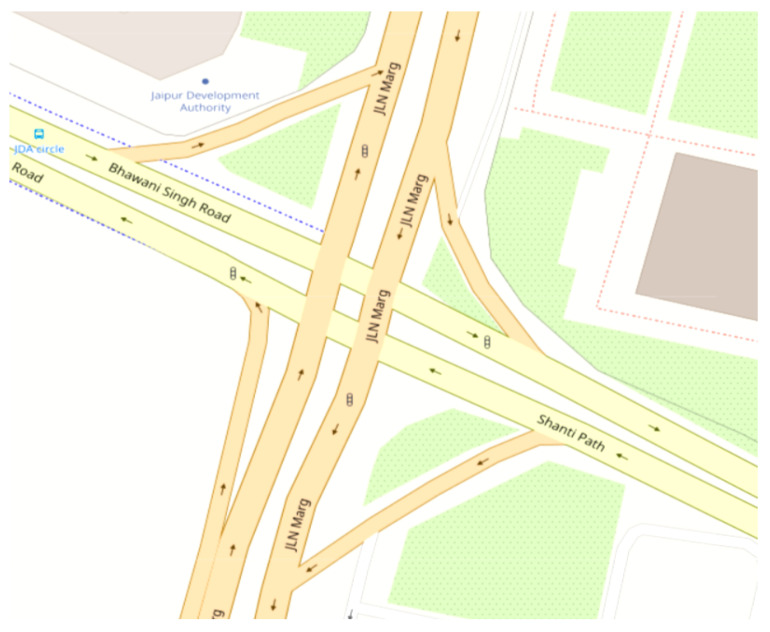
Road segment (Jaipur JLN Marg, India).

**Figure 10 sensors-24-02334-f010:**
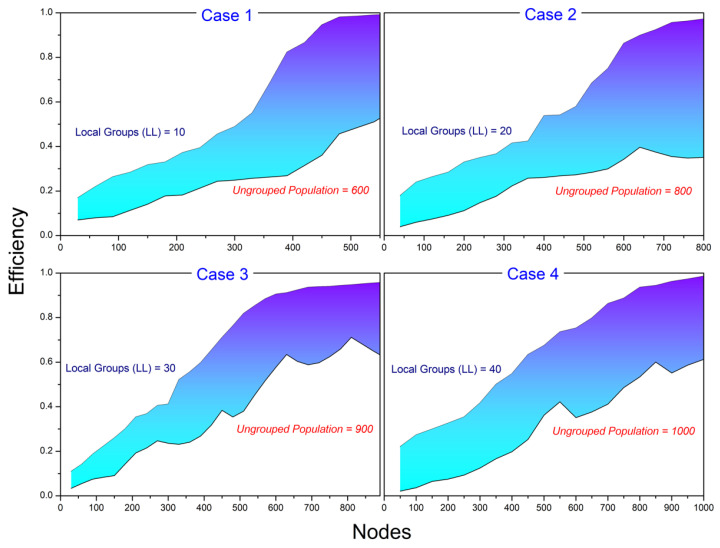
w-SMNO efficiency.

**Figure 11 sensors-24-02334-f011:**
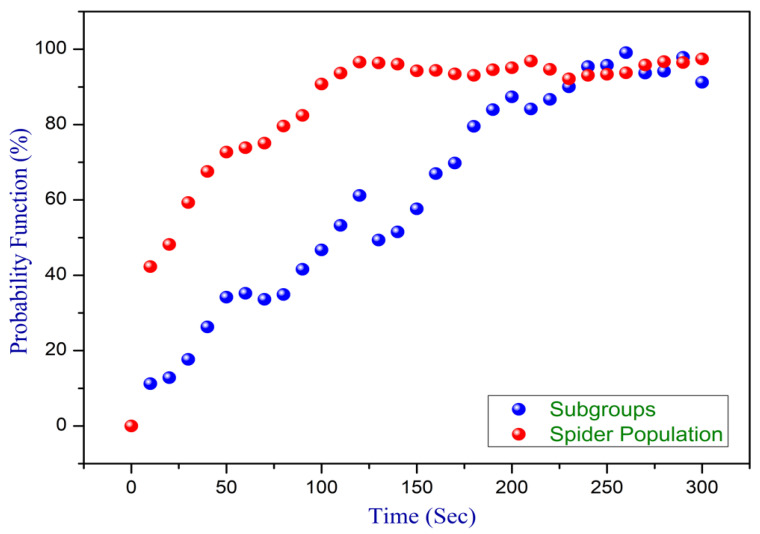
Experiment 1: probability analysis for LL with population size.

**Figure 12 sensors-24-02334-f012:**
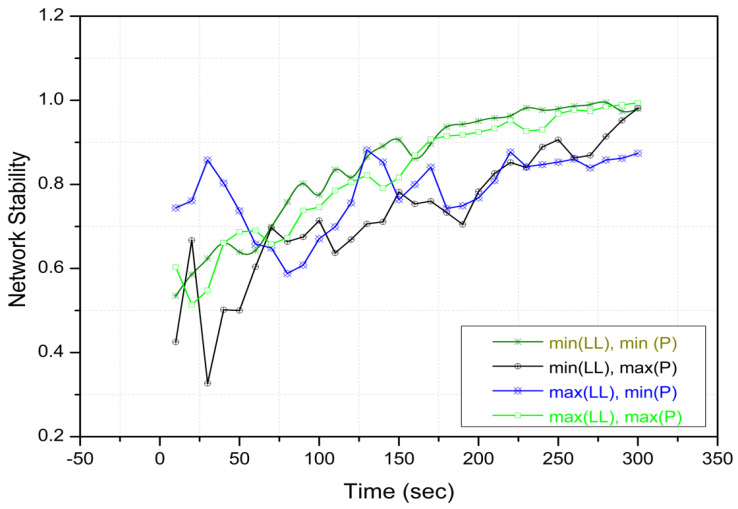
Experiment 2: w−SMNO network stability under (min, max) group.

**Figure 13 sensors-24-02334-f013:**
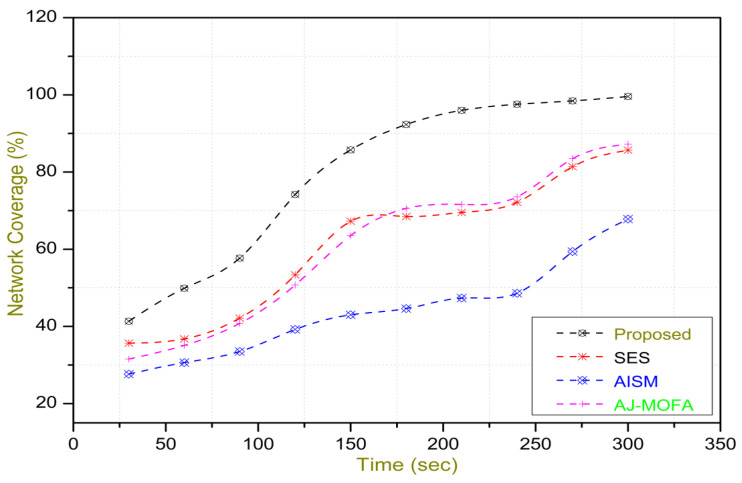
Coverage with time.

**Figure 14 sensors-24-02334-f014:**
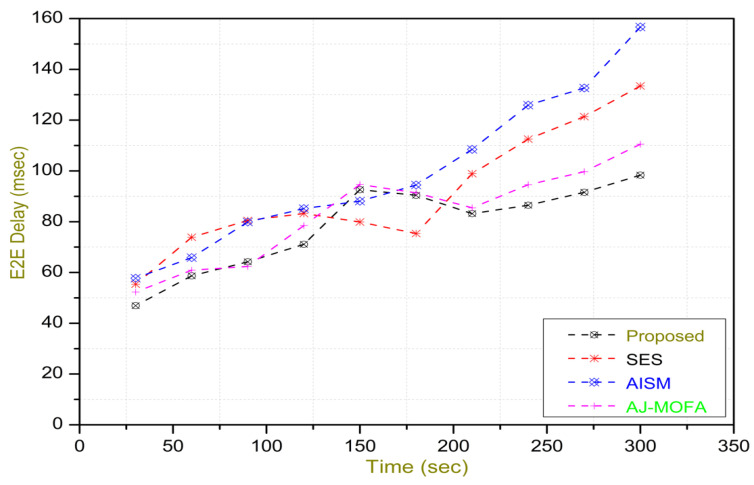
E2E delay with time.

**Figure 15 sensors-24-02334-f015:**
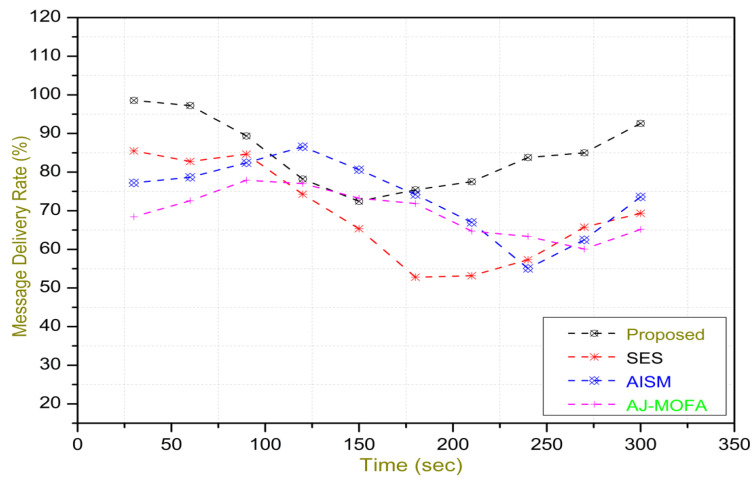
MDR with time.

**Figure 16 sensors-24-02334-f016:**
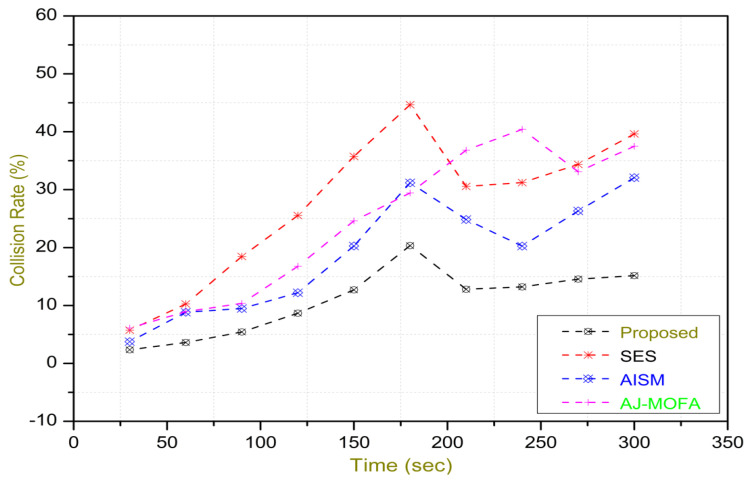
Collision—rate with time.

**Figure 17 sensors-24-02334-f017:**
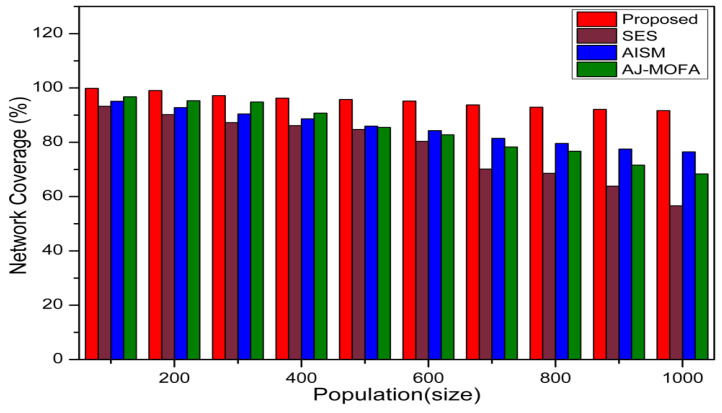
Coverage with population.

**Figure 18 sensors-24-02334-f018:**
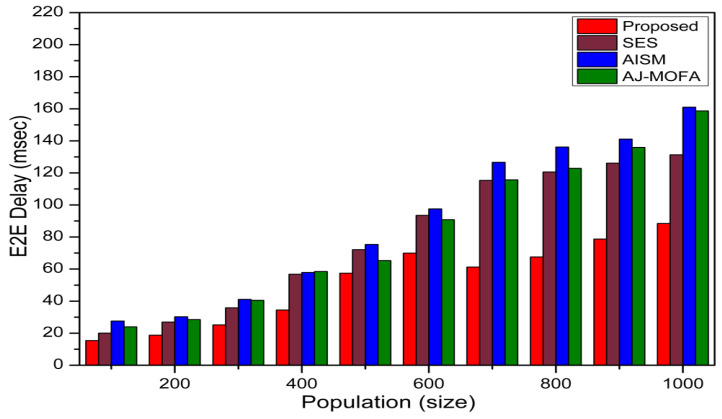
E2E delay with population.

**Figure 19 sensors-24-02334-f019:**
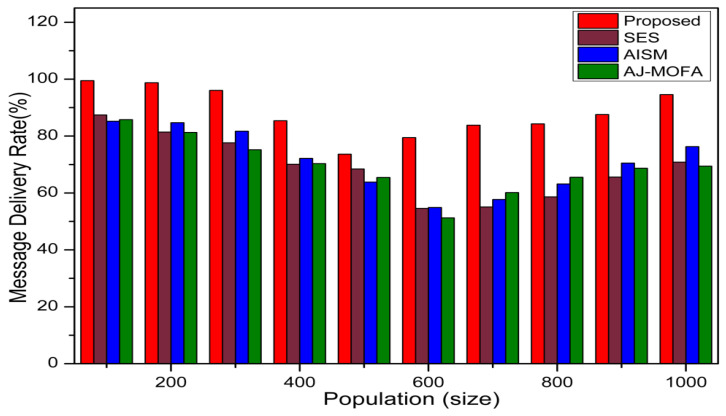
MDR with population.

**Figure 20 sensors-24-02334-f020:**
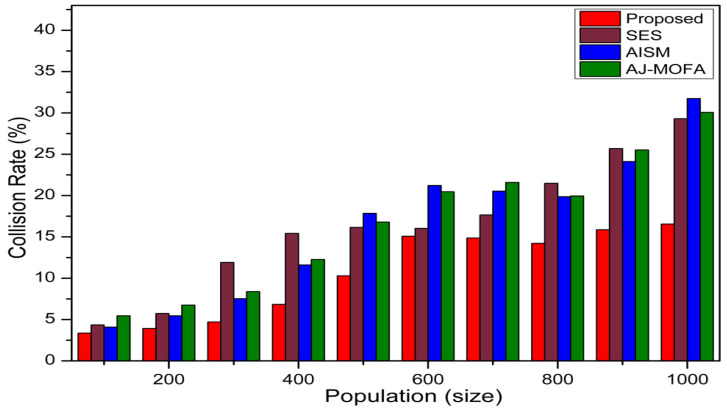
Collision—rate with population.

**Figure 21 sensors-24-02334-f021:**
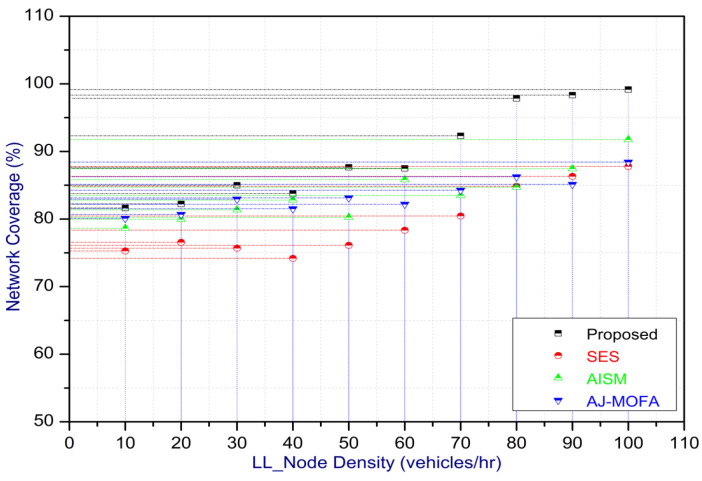
Coverage with LL node density.

**Figure 22 sensors-24-02334-f022:**
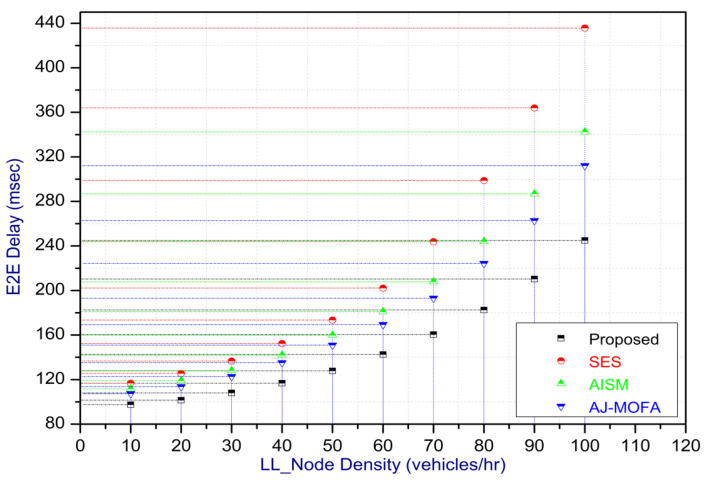
E2E delay with LL node density.

**Figure 23 sensors-24-02334-f023:**
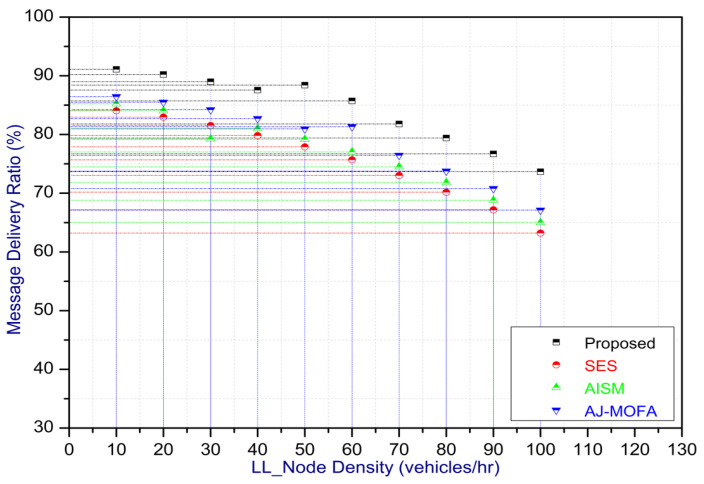
MDR with LL node density.

**Figure 24 sensors-24-02334-f024:**
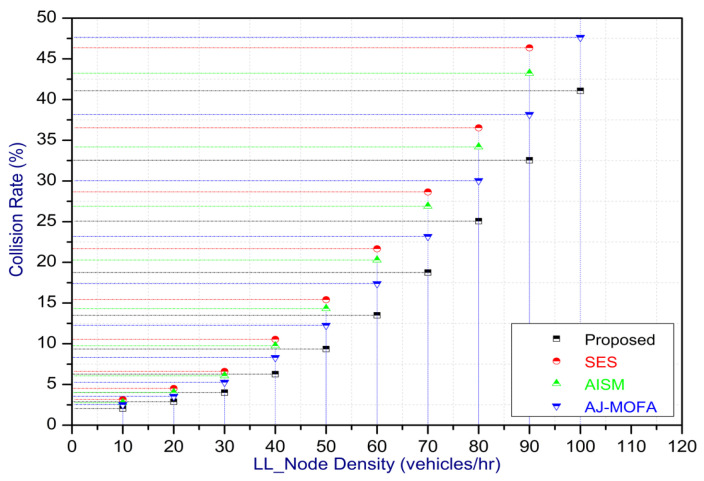
Collision—rate with LL node density.

**Table 1 sensors-24-02334-t001:** Threshold instances (LL group).

Instances		Group (Order of Score)	Relay Node
Group Population	Neighbors (Qth Group LL)		
($P_{LL}{)}_{Q}>=P_{Threshold}$	($N_{LL}{)}_{Q}>=LL_{Threshold}$	LL_*Q*_	Maximum weight node (Machine learning model)
($P_{LL}{)}_{Q}<P_{Threshold}$		LL_*Q*_	
($P_{LL}{)}_{Q}>=P_{Threshold}$	($N_{LL}{)}_{Q}<LL_{Threshold}$	LL_*Q*_	
($P_{LL}{)}_{Q}<P_{Threshold}$		LL_$Q^{\prime }$_	

**Table 2 sensors-24-02334-t002:** Comparison analysis between the SES, AISM, AJ-MOFA, and w-SMNO protocols.

Protocols	SES	AISM	AJ-MOFA	Proposed w-SMNO
**VANET domain**	Routing	Routing	Routing	Routing
**Optimization strategy**	Social spider	Ant colony	MO firefly	Spider monkey
**Sub-strategy**	Clustering	None	Clustering	ML model
**Source identification**	Position-based	Position-based	Position-based	Position-based
**Route configuration**	Yes	Yes	Yes	Yes
**Route selection**	Yes	Yes	Yes	Yes
**Disconnection rate**	Medium	Medium	Low	Low
**Route selection approach**	Vehicle link	Metrics	Pareto method	Weighted metrics
**Vehicle density (Max)**	100	300	100	100
**Network connection**	V2V	V2I	V2V	V2V
**Comparison**	CLPSO, GA, ACO	GeoSVR, RAGR, EGSR	CPB, NSGA-II	SES, AISM, AJ-MOFA
**Environment**	Urban	Urban	Urban	Urban
**Simulation tool**	MATLAB	NS2.34	MATLAB	NS-2.35

**Table 3 sensors-24-02334-t003:** Simulation parameters.

Parameters	Values
Population Size	1000
Local Groups (LL)	[5–40]
Group Population	[30–60]
Test Time	300 seconds
Traffic	CBR
Number of Tests	10
Region	[500 × 500] m^2^
Frame Length	2 Kbytes
Displacement	Random
Mobility	RWP
Node Speed	[20–100] km/h
Transport Protocol	UDP
Transmission	300 m
MAC	IEEE 802.11b
Propagation	2-Way
Routing Protocol	OLSR
Experiment Tool	NS 2.35

**Table 4 sensors-24-02334-t004:** w-SMNO scenarios carrying non-grouped conventional vehicle density.

State	w-SMNO		Non-Grouped Conventional Vehicle Density
	Local Groups (Size)	Population (Individual LL)	
Case 1	10	60	600
Case 2	20	40	800
Case 3	30	30	900
Case 4	40	25	1000

**Table 5 sensors-24-02334-t005:** Network stability under (min, max) group.

Category	Spider Monkey (Vehicles)	
	Groups (LL)	Population (P)
A (min, min)	10	600
B (min, max)	10	1000
C (max, min)	40	600
D (max, max)	40	1000

**Table 6 sensors-24-02334-t006:** Simulation results proposed for delay and network coverage.

Factors		E2E Delay (msec)				Network Coverage (%)			
State-of-the-Art		Proposed	SES	AISM	AJ-MOFA	Proposed	SES	AISM	AJ-MOFA
**Time (s)**	30	46.93	55.36	57.73	52.29	41.35	35.66	27.63	31.54
	60	58.67	73.9	65.86	60.9	49.88	36.76	30.61	35.08
	90	64.19	80.5	79.86	62.38	57.69	42.13	33.49	40.77
	120	71.09	83.22	85.17	78.47	74.18	53.37	39.22	50.71
	150	92.63	79.86	88.11	94.65	85.74	67.28	43	63.52
	180	90.43	75.36	94.45	91.37	92.33	68.47	44.62	70.57
	210	83.2	98.93	108.48	85.44	96	69.53	47.36	71.62
	240	86.47	112.57	125.93	94.58	97.58	72.15	48.58	73.55
	270	91.58	121.36	132.67	99.72	98.44	81.39	59.41	83.46
	300	98.36	133.49	156.71	110.51	99.58	85.66	67.84	87.23
**Population (Count)**	100	15.37	20.07	27.56	23.93	99.87	93.27	95.12	96.75
	200	18.73	26.96	30.24	28.47	99.04	90.18	92.77	95.3
	300	25.19	35.84	41.06	40.52	97.19	87.26	90.43	94.83
	400	34.46	56.76	57.87	58.45	96.24	86.14	88.62	90.74
	500	57.44	72.08	75.34	65.29	95.76	84.7	85.94	85.47
	600	69.95	93.53	97.55	90.8	95.17	80.35	84.27	82.77
	700	61.25	115.29	126.64	115.62	93.78	70.12	81.46	78.26
	800	67.56	120.57	136.14	122.84	92.92	68.56	79.56	76.68
	900	78.69	126.12	141.08	135.88	92.11	63.84	77.51	71.59
	1000	88.48	131.26	161	158.65	91.64	56.61	76.48	68.34
**Local Leader Density**	10	97.44	116.84	111.71	107.27	81.63	75.28	78.63	80.09
	20	101.56	125.51	119	113.68	82.24	76.564	79.958	80.678
	30	108.02	136.7	128.08	122.67	85.006	75.682	81.347	82.956
	40	116.77	152.34	141.87	135.12	83.78	74.192	82.813	81.518
**Local Leader Density**	50	127.85	173.46	159.79	150.99	87.66	76.107	80.251	83.127
	60	142.6	202.15	180.97	169.41	87.49	78.348	85.813	82.195
	70	160.43	243.82	207.85	193.15	92.319	80.443	83.477	84.264
	80	182.49	298.78	244.02	224.37	97.861	84.837	84.698	86.236
	90	210.38	364.02	286.83	262.8	98.332	86.304	87.435	85.117
	100	244.93	435.66	342.5	312.08	99.157	87.774	91.738	88.421

**Table 7 sensors-24-02334-t007:** Simulations result of proposed for message delivery rate and collision rate.

Factors		Message Delivery Rate (%)				Collision Rate (%)			
State-of-the-Art		Proposed	SES	AISM	AJ-MOFA	Proposed	SES	AISM	AJ-MOFA
**Time (s)**	30	98.56	85.49	77.21	68.43	2.35	5.73	3.76	6.11
	60	97.24	82.76	78.66	72.59	3.61	10.26	8.83	8.95
	90	89.37	84.64	82.42	77.91	5.42	18.46	9.48	10.37
	120	78.19	74.28	86.57	77.02	8.68	25.54	12.2	16.77
	150	72.43	65.36	80.62	73.25	12.7	35.76	20.27	24.62
	180	75.41	52.81	74.18	71.84	20.35	44.68	31.18	29.43
	210	77.52	53.16	67	64.73	12.8	30.57	24.84	36.82
	240	83.77	57.33	55.07	63.34	13.21	31.22	20.26	40.41
	270	85	65.74	62.46	60.08	14.55	34.37	26.34	33.08
	300	92.6	69.35	73.58	65.16	15.16	39.63	32.08	37.51
**Population (Count)**	100	99.48	87.44	85.2	85.76	3.36	4.35	4.08	5.45
	200	98.75	81.43	84.68	81.28	3.92	5.73	5.46	6.75
	300	96.07	77.6	81.7	75.19	4.71	11.91	7.52	8.39
	400	85.4	70.09	72.16	70.32	6.84	15.43	11.61	12.27
**Population (Count)**	500	73.63	68.42	63.84	65.46	10.29	16.15	17.85	16.8
	600	79.48	54.59	54.91	51.27	15.08	16.03	21.22	20.46
	700	83.8	55.1	57.71	60.14	14.86	17.66	20.53	21.59
	800	84.28	58.64	63.18	65.51	14.22	21.49	19.87	19.95
	900	87.56	65.58	70.5	68.7	15.87	25.68	24.11	25.52
	1000	94.57	70.84	76.29	69.45	16.55	29.3	31.73	30.06
**Local Leader Density**	10	91.09	84.06	85.23	86.45	2.043	3.153	2.761	2.523
	20	90.24	82.96	84.03	85.5	2.883	4.513	4.011	3.543
	30	89	81.53	79.24	84.21	4.003	6.603	6.071	5.263
	40	87.56	79.82	80.99	82.71	6.273	10.543	9.751	8.313
	50	88.42	77.89	79.19	80.96	9.353	15.413	14.321	12.253
	60	85.73	75.68	77.03	81.33	13.493	21.673	20.291	17.373
	70	81.81	73.02	74.5	76.45	18.753	28.653	26.901	23.183
	80	79.41	70.18	71.8	73.77	25.073	36.523	34.191	30.043
	90	76.7	67.18	68.8	70.8	32.543	46.363	43.231	38.173
	100	73.66	63.21	65.01	67.12	41.073	57.633	53.641	47.633

## Data Availability

On justified request, the corresponding author may share relevant data and materials used in the manuscript. The Network Simulator software with version 2.35 downloaded from www.nsnam.org is used to develop and run the algorithmic code.

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
