# Peer review of "A Novel Spider Monkey Optimization for Reliable Data Dissemination in VANETs Based on Machine Learning"

_sensors, 2024, doi:10.3390/s24072334_

Round 1

Reviewer 1 Report (Previous Reviewer 2)

Comments and Suggestions for Authors

The authors have taken into account all the recommendations I had made, and for the one point on which they disagreed, they have provided a clear and logical justification. This article seems to me to be acceptable in its present form.

Author Response

Reviewer 2 Report (Previous Reviewer 3)

Comments and Suggestions for Authors

In the manuscript, the authors introduce a novel spider monkey based optimization method to generate efficient relays VANETs. Overall, the paper is fine. My main comments are given as follows:

1.   What is the motivation behind the research?

2.    I think mathematical representation/model should not be considered as a contribution.

3.    Novelty of the paper needs to be strengthened, as the methods employed lack innovation or improvement.

4.    Has leader selection taken into account the dynamic characteristics of VANETs? Such hierarchical network structures may pose additional security risks, as they cannot guarantee the compliance of leader behavior.

5.    In the proposed model, the process of leader selection can be abstracted as a mobile assignment problem. The authors can refer to some similar works in ACM Transactions on Sensor Networks and IEEE Transactions on Mobile Computing with key words, e.g., multiple mobile chargers, minimal wait-for delay, header selection.

6.    The authors only compare the performance of the proposed method with simulation results in NS-2, which can not demonstrate the advantage of the proposed method. It may be much better if authors can specify the simulation results.

7.    It seems the authors are unaware the recent trends in vehicle communication. Must refer recent journals/citations in 2022, 2023, 2024 years.

Comments on the Quality of English Language

Should be improved further.

Author Response

This manuscript is a resubmission of an earlier submission. The following is a list of the peer review reports and author responses from that submission.

Round 1

Reviewer 1 Report

Comments and Suggestions for Authors

The paper talks about a networked connected vehicles systems and their optimization. I feel this particular application is too futuristic and I can not think of any theoretical guarantee in the paper, to prove the applicability of the proposed method.

Comments on the Quality of English Language

The paper needs a thorough grammatical editing. 

Author Response

Please see the attachmnet.

Reviewer 2 Report

Comments and Suggestions for Authors

This article is well written and proposes an interesting approach to the optimization of communications in VANETs. However, here are a few remarks that I think should be taken into account to improve the quality of this article:

1) In the introduction, the environment under consideration should be better highlighted. Indeed, depending on the article, different scenarios can be considered for VANETs (with/without infra) and it would seem interesting to me to better target what is intended here. 

2) a reduction in the number of bullets for the contributions in this article. Different points such as 2,3 and 4,5 would seem to me to need to be put together

3) the references mentioned in the Related Works section are numerous. However, their contributions/limitations seem to me to be underemphasized. Moreover, I wonder whether these are always the most relevant references. The number of references relating to Machine Learning/Deep Learning, an integral part of this article, remains very limited, despite the fact that a great deal of work has already been proposed in the literature.

4) I wonder whether the first page of section 3 is necessary. Wouldn't it be more relevant to apply the solution directly to the VANET environment? Is there any real added value in representing the whole solution in this way?

5) The quality of the figures in section 3 seems to me to be very limited and needs to be improved.

6) The equations presented in sections 3.1 and 3.2 seem to me to remain very limited and to make a very high level of abstraction on the quality of the communication links. Wouldn't a higher level of precision be relevant here?

7) The evaluations carried out to measure the solution's level of performance seem relevant to me. However, it would seem essential to me to rework this section to make it more fluid and impactful, notably by grouping the figures together where relevant, improving the quality of some of them (e.g. Figure 9) and better organizing the text associated with the results in this section. 

Reviewer 3 Report

Comments and Suggestions for Authors

This paper proposes a novel machine learning approach based on spider monkey optimization for reliable data dissemination in vehicular ad hoc networks (VANETs). The approach, called weighted estimated Spider Monkey-based Nature-Inspired Optimization (w-SMNO), aims to reduce communication delays and improve system accuracy by identifying efficient relay nodes. Simulation results demonstrate significant improvements in coverage, end-to-end delay, message delivery rate, and collision rate compared to existing approaches.

Consider incorporating specific comparative tables or charts in the text to visually illustrate the characteristics and performance differences among various methods. Additionally, highlight the innovative aspects of the proposed method, particularly its advancements compared to existing approaches. It may be beneficial to broaden the scope of experimental validation by including more diverse scenarios, such as varying vehicle densities and communication environments, to comprehensively evaluate the proposed methods' performance and applicability. Enhance the interpretation and analysis of experimental results, providing detailed descriptions of observed trends and explanations for the outcomes, enabling readers to grasp the underlying mechanisms and influencing factors.

Comments on the Quality of English Language

Quality of English Language can be improved.
